# Computing Committor Functions for the Study of Rare Events Using Deep Learning with Importance Sampling

## Abstract

The committor function is a central object of study in understanding transitions between metastable states in complex systems. However, computing the committor function for realistic systems at low temperatures is a challenging task, due to the curse of dimensionality and the scarcity of transition data. In this paper, we introduce a computational approach that overcomes these issues and achieves good performance on complex benchmark problems with rough energy landscapes. The new approach combines deep learning, importance sampling and feature engineering techniques. This establishes an alternative practical method for studying rare transition events among metastable states of complex, high dimensional systems.

## 1 Introduction

Understanding transition events between metastable states is of great importance in the applied sciences. Well-known examples of the transition events include nucleation events during phase transitions, conformational changes of bio-molecules, dislocation dynamics in crystalline solids, etc. The long time scale associated with these events is a consequence of the disparity between the effective thermal energy and typical energy barrier of the systems. The dynamics proceeds by long waiting periods around metastable states followed by sudden jumps from one state to another. For this reason, the transition event is called rare event. The main objective in the study of rare events is to understand the transition mechanism, such as the transition pathway and transition states (or bottlenecks). Some numerical methods have been proposed for this purpose, among which the well-known ones include the nudged elastic band method (Jónsson et al., 1998), the string method (E et al., 2002; 2007; Ren et al., 2005), the action based method (Olender & Elber, 1996), and the transition path sampling technique (Bolhuis et al., 2002; Dellago et al., 2002), accelerated molecular dynamics (Voter, 1997), etc.

One object that plays an important role in understanding the transition event is the committor function. This is a function which is defined in the configuration or phase space and describes the progress of the transition. Most of the interesting information regarding the transition can be extracted from the committor function (E et al., 2005; Ren et al., 2005; E & Vanden-Eijnden, 2010). For example, the transition states lie in the region where the committor value is around $1/2$; the transition path ensemble and the transition rate can be computed as well, based on the committor function and the equilibrium probability distribution of the system.

The committor function has a very simple mathematical description - it satisfies the backward Kolmogorov equation. However, it is very difficult to compute in practice, due to the curse of dimensionality. For example, consider a bio-molecule consisting of $N$ atoms, then the equation needs be solved in the configuration space of dimension $3N$. Traditional numerical approaches, such as finite difference or finite element methods, are computationally prohibited even when $N = 2$ or 3, and obviously out of the question for realistic physical systems. Alternative methods have been proposed. In the transition path sampling technique (Bolhuis et al., 2002; Dellago et al., 2002), the committor function is computed using Monte Carlo methods. In the finite-temperature string method (Ren et al., 2005), the problem is reformulated as an equivalent variational form, which is minimized in a particular function space. In the work of Lai & Lu (2018), the Kolmogorov equation for the committor function is solved using a point cloud discretization.

More recently, it is proposed to compute the committor function using neural networks (Khoo et al., 2018). They obtained satisfactory numerical results for a model problem in 10 dimensions. The success of this approach hinges on the availability of data, especially in the transition state region where the committor function changes sharply from 0 to 1. As the term "rare event" suggests, such data is rarely available. In Khoo et al. (2018), the data was generated by solving the underlying Langevin dynamics at the physical temperature. While this works well when the physical temperature is high in which case transitions are easily observed, it becomes less efficient when the temperature is low, i.e. the case of rare events.

In this work, we propose an importance sampling technique to overcome this difficulty. Specifically, we generate the data from the Langevin dynamics at an artificial temperature. This temperature is high enough so that the transition event between the metastable states becomes less rare or even frequent. This enables us to collect sufficient amount of data in the transition state region. The difference between the physical temperature and the artificial temperature is accounted for by the likelihood ratio in the objective function to be minimized. Furthermore, we introduce a new method to impose the boundary conditions for the committor function, and employ collective variables as the input features for the neural network.

The rest of the paper is organized as follows. The background and problem formulation is discussed in Section 2. In Section 3, we introduce the key ingredients of our method, including the method of imposing the Dirichlet boundary conditions, the importance sampling technique, and the neural network architecture. Numerical results for two examples are presented in Section 4. Finally, we draw conclusions in Section 5.

## 2 BACKGROUND

The typical starting point of transition modeling is the specification of a potential energy function $V : \Omega \subset \mathbb{R}^n \to \mathbb{R}$, which takes as inputs the microscopic configuration $x$ of the system (e.g. the positions of the constituent atoms of a bio-molecule). The subset $\Omega$ is the configuration space under consideration. Through standard statistical mechanics arguments, the probability density of the system's configuration is determined by the potential energy function $V$ via the Boltzmann-Gibbs distribution

$$p(x) = \frac{1}{\mathcal{Z}} e^{-\beta V(x)}, \tag{1}$$

where $\beta = 1/k_B T$ is the inverse temperature and $\mathcal{Z} = \int_\Omega e^{-\beta V(x)} dx$ is the normalization factor, or partition function.

To study dynamical properties, one may consider the noisy gradient flow induced by $V$ in the form of an over-damped Langevin equation:

$$\dot{x}(t) = -\nabla V(x(t)) + \sqrt{2\beta^{-1}} \eta(t), \tag{2}$$

where $\eta$ is a white noise. One can check that (2) has invariant distribution (1). Through this dynamical model, it is clear how one can then introduce the notion of stability of the system: local minima, or a collection of local minima, of $V$ correspond to metastable configurations; at low temperatures (large $\beta$), the system will remain in these configurations for exponentially long times before making transition to another such configuration. Our principal goal is to study the transition dynamics between two metastable configurations.

Consider two distinct metastable regions $A, B \subset \Omega$, $A \cap B = \varnothing$. Consider the first hitting times of set $A$ and $B$

$$\tau_A(x) = \inf\{t \geq 0 : x(t) \in A, x(0) = x\}, \qquad \tau_B(x) = \inf\{t \geq 0 : x(t) \in B, x(0) = x\}. \tag{3}$$

The committor function $q : \Omega \to [0, 1]$ is defined by

$$q(x) = \mathrm{Prob}\{\tau_B(x) < \tau_A(x)\}. \tag{4}$$

It can be shown that much of the vital information regarding transition pathways and rates can be extracted from the committor function (E et al., 2005; E & Vanden-Eijnden, 2010). Hence, an efficient numerical method to compute it for general systems is an important problem in computational chemistry, structural biology and materials science.

We consider the system in the domain $\Omega \setminus (A \cup B)$ since the definition of $q$ involves the first hitting times of the boundary of $A$ and $B$. It can be shown that $q$ satisfies the backward Kolmogorov equation with Dirichlet boundary condition (Gardiner, 1985),

$$\begin{cases} \nabla V \cdot \nabla q - \beta^{-1} \Delta q = 0, & x \in \Omega \setminus (A \cup B), \\ q(x) = 0, \ x \in \partial A; \quad q(x) = 1, \ x \in \partial B, \end{cases} \tag{5}$$

where $\nabla = (\frac{\partial}{\partial x_1}, ..., \frac{\partial}{\partial x_n})$ is the gradient operator, $\Delta = \sum_{i=1}^{n} \frac{\partial^2}{\partial x_i^2}$ is the Laplace operator and $\partial A$ ($\partial B$) denotes the boundary of $A$ ($B$). In addition, we impose the boundary condition $\nabla q \cdot \mathbf{n} = 0$ on $\partial \Omega$. The committor function is also the solution to the variational problem

$$\min_{q} \frac{1}{Z} \int_{\Omega \setminus (A \cup B)} |\nabla q(x)|^2 e^{-\beta V(x)} dx, \tag{6}$$

where $Z = \int_{\Omega \setminus (A \cup B)} e^{-\beta V(x)} dx$, subject to the condition

$$q(x) = 0, \ x \in \partial A; \quad q(x) = 1, \ x \in \partial B. \tag{7}$$

The key challenge in using (5) to compute $q$ is the curse of dimensionality: as the dimension $n$ of the configuration $x$ increases, the computational complexity of classical finite difference and finite element methods increase exponentially, thereby prohibiting the use of these methods for realistic models. Although the variational formulation somewhat ameliorate this issue, at low temperatures (large $\beta$), which is often the regime of interest for studying rare transition events, solving (6) becomes more challenging as the measure $Z^{-1} e^{-\beta V}$ becomes more "singular".

In the next section, we introduce our proposed method of combining deep learning and importance sampling to efficiently compute $q$ for high-dimensional systems and at low temperatures.

## 3 METHODS

Based on the approach introduced in Khoo et al. (2018), we use a deep neural network $q_\theta$, where $\{\theta\}$ are trainable variables, to parameterize the committor function. Note that since the range of the committor function is $[0, 1]$, we use a sigmoid activation for the output layer. With this parameterization scheme, the variational problem (6) can be posed as an unsupervised learning problem

$$\arg\min_{\theta} \int_{\Omega \setminus (A \cup B)} |\nabla q_\theta(x)|^2 p_\beta(x) dx,$$
$$q_\theta(x) = 0, \ x \in \partial A; \quad q_\theta(x) = 1, \ x \in \partial B. \tag{8}$$

with $p_\beta(x) = Z^{-1} e^{-\beta V(x)}$ being the equilibrium distribution. To solve (8), training data following the equilibrium distribution $p_\beta$ are sampled by taking snapshots from long trajectory of the Langevin dynamics (2),

Below, we outline the three main novel aspects of our approach: 1) a network structure that deals naturally with the boundary condition (7); 2) an importance sampling technique that allows for the efficient computation of committor functions at low temperatures, which is important for studying transition events; and 3) Using collective variables as a form of crude feature selection to improve the efficiency of learning.

### 3.1 BOUNDARY CONDITIONS

To handle boundary conditions, Khoo et al. (2018) proposed adding a penalty term enforcing (7) into the functional to be minimized to obtain the augmented problem

$$\arg\min_{\theta} \frac{1}{Z} \int_{\Omega \setminus (A \cup B)} |\nabla q_\theta(x)|^2 e^{-\beta V(x)} dx + \lambda \int_{\partial A} q_\theta^2(x) d\mu_{\partial A}(x) + \lambda \int_{\partial B} (1 - q_\theta(x))^2 d\mu_{\partial B}(x), \tag{9}$$

where $\lambda$ is a parameter controlling the magnitude of the penalty terms, $\mu_{\partial A}$ and $\mu_{\partial B}$ are the equilibrium distribution restricted on $\partial A$ and $\partial B$, respectively.

While this is straightforward to implement, the introduction of an additional penalty parameter may require more tuning. Instead, we proceed differently by introducing a different parameterization of $q_\theta$ that naturally satisfies the boundary conditions (7). Concretely, we consider the composite form

$$q(x) = (1 - \chi_A(x)) \left[ (1 - \chi_B(x))\tilde{q}(x) + \chi_B(x) \right], \quad x \in \Omega \setminus (A \cup B). \tag{10}$$

Here $\chi_A(x)$ and $\chi_B(x)$ are two given smooth functions such that

$$\chi_A(x) = \begin{cases} 1, & x \in \partial A \\ 0, & x \in \Omega \setminus (A^\epsilon \cup B^\epsilon), \end{cases} \qquad \chi_B(x) = \begin{cases} 1, & x \in \partial B \\ 0, & x \in \Omega \setminus (A^\epsilon \cup B^\epsilon), \end{cases} \tag{11}$$

where $A^\epsilon$ and $B^\epsilon$ are two sets expanded from $A$ and $B$,

$$A^\epsilon = \left\{ x \in \Omega : \inf_{y \in A} |x - y| \leq \epsilon \right\}, \quad B^\epsilon = \left\{ x \in \Omega : \inf_{y \in B} |x - y| \leq \epsilon \right\}. \tag{12}$$

Away from $\partial A$ and $\partial B$, $\chi_A$ and $\chi_B$ changes smoothly from 1 to 0 in a region of width $\epsilon$, respectively. The function $q(x)$ of the form in (10) strictly satisfies the boundary conditions (7) and agrees with $\tilde{q}$ outside $A^\epsilon$ and $B^\epsilon$,

$$q(x) = \tilde{q}(x), \quad \text{for } x \in \Omega \setminus (A^\epsilon \cup B^\epsilon). \tag{13}$$

We then use a deep neural network to parameterize the function $\tilde{q}$, hence approximating the committor function $q(x)$ as

$$q_\theta(x) = (1 - \chi_A(x)) \left[ (1 - \chi_B(x))\tilde{q}_\theta(x) + \chi_B(x) \right], \quad x \in \Omega \setminus (A \cup B), \tag{14}$$

where $\{\theta\}$ are trainable variables of the neural network. Thus, the constrained minimization problem (8) reduces to an unconstrained problem

$$\arg\min_\theta \frac{1}{Z} \int_{\Omega \setminus (A \cup B)} |\nabla_x q_\theta(x)|^2 e^{-\beta V(x)} dx, \tag{15}$$

where $q_\theta(x)$ takes the form (14). Consequently, only the data following the stochastic process (2) is needed to train the neural network.

## 3.2 Importance Sampling

Another challenging problem is the sampling of $p_\beta$ at low temperature $T$ (large $\beta$). Naive sampling using the dynamics (2) at low temperatures yields very few transition events and hence the distribution $p_\beta$ is not explored well enough to solve (15). Specifically, with very few observed transition events, the majority of the sampled data will be clustered near the metastable states $A$ and $B$, with very few distributed in the region of our interest, i.e. the transition state region which lies in between $A$ and $B$. This will lead to a poor estimate of the committor function.

To address this problem, we consider another form of the problem (15)

$$\arg\min_\theta \frac{1}{Z'} \int_{\Omega \setminus (A \cup B)} |\nabla_x q_\theta(x)|^2 e^{-(\beta - \beta')V(x)} e^{-\beta' V(x)} dx, \tag{16}$$

where $Z' = \int_{\Omega \setminus (A \cup B)} e^{-\beta' V(x)} dx$ is the normalization factor and $\beta' = 1/k_B T'$, with $T' > T$.

Solving the new problem (16) requires sampling of $p_{\beta'}(x) = Z'^{-1} e^{-\beta' V(x)}$ by simulating the Langevin dynamics (2) at the high temperature $T'$. This becomes much more efficient when $T'$ is sufficiently large, as energy barriers are much easier to cross at higher temperatures. In essence, this is a form of importance sampling, and the function $e^{-(\beta - \beta')V(x)}$ is the likelihood ratio associated with the change of measure. Note that the choice of importance sampling measure $p_{\beta'}$ is not arbitrary. Its purpose is to produce enough sample points in the *a priori* unknown transition region for us to estimate the committor function at low temperature $T$. Generic sampling measures (e.g. uniform) will not achieve this goal in moderately high dimensions.

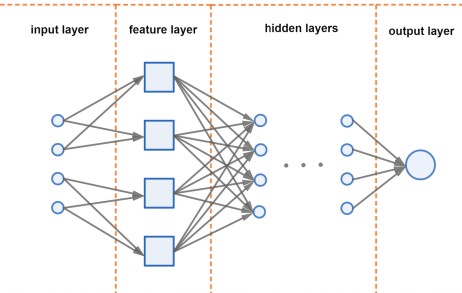

Figure 1: Diagram of neural network structure with feature layer. The feature layer consists of collective variables information. The connection between the input and feature layers is determined by the definition of the collective variables, whereas the feature layer and the first hidden layer are fully connected.

### 3.3 COLLECTIVE VARIABLES

Very often, only a few coarse-grained variables, called the collective variables and denoted by $(z_1(x), z_2(x), ..., z_m(x))$ with $m \ll n$, play a major role in the transition event. For example, in conformational changes of bio-molecules, it is often that only a few torsion or bond angles participate the transitions between metastable states. In this situation, the committor function, being the reaction coordinate for the transition, depends only on these collective variables

$$q(x) = f(z_1(x), ..., z_m(x)). \tag{17}$$

This is a form of dimensional reduction or feature engineering, and the choice of $\{z_i\}$ often allows one to build some degree of physical knowledge into the parameterization of $q$.

For the alanine dipeptide example to be discussed in the next section, we shall make use of collective variables in the design of the first input transformation layer. The architecture of the network is summarized in Figure 1.

## 4 NUMERICAL RESULTS

We demonstrate the effectiveness of the proposed numerical method using two benchmark problems: one is the Mueller potential extended to high dimensions, the other is the isomerization of alanine dipeptide.

### 4.1 EXTENDED MUELLER POTENTIAL

We first consider the Mueller potential embedded in the 10-dimensional (10D) space (Khoo et al., 2018),

$$V(x) = V_m(x_1, x_2) + \frac{1}{2\sigma^2} \sum_{i=3}^{10} x_i^2, \quad x \in \mathbb{R}^{10} \tag{18}$$

where $V_m(x_1, x_2)$ is the rugged Mueller potential in two dimensions (2D),

$$V_m(x_1, x_2) = \sum_{i=1}^{4} D_i \exp\left(a_i(x_1 - X_i)^2 + b_i(x_1 - X_i)(x_2 - Y_i) + c_i(x_2 - Y_i)^2\right)$$
$$+ \gamma \sin(2k\pi x_1)\sin(2k\pi x_2), \tag{19}$$

and a harmonic potential is used in each of the other eight dimensions. The parameters $\gamma$ and $k$ control the roughness of the energy landscape, and $\sigma$ controls the scale of the quadratic terms. In this example, we use $\gamma = 9$, $k = 5$, $\sigma = 0.05$. The other parameters are taken from Lai & Lu (2018).

We first use the finite element method (FEM) to solve the backward Kolmogorov equation (5) for the 2D rugged Mueller potential $V_m$ on $\tilde{\Omega} = [-1.5, 1] \times [-0.5, 2]$ by the solver *FreeFem++* (Hecht,

Table 1: Comparison of the committor function $q_\theta$ obtained using neural network with the FEM solution $q$ for the extended Mueller potential. The error is computed on the 100 transition states sampled from the dynamics (21). The statistics of the error (mean $\pm$ deviation) is based on 10 independent computations.

| size of data set | network structure | RMSE | MAE |
|---|---|---|---|
| $2 \times 10^5$ | 10-20-1 | $0.0464 \pm 0.0179$ | $0.0372 \pm 0.0151$ |
| | 10-20-20-1 | $0.0483 \pm 0.0106$ | $0.0390 \pm 0.0102$ |
| | 10-20-20-20-1 | $0.0483 \pm 0.0170$ | $0.0390 \pm 0.0150$ |
| $5 \times 10^5$ | 10-20-1 | $0.0331 \pm 0.0096$ | $0.0282 \pm 0.0098$ |
| | 10-20-20-1 | $0.0363 \pm 0.0113$ | $0.0307 \pm 0.0116$ |
| | 10-20-20-20-1 | $0.0368 \pm 0.0108$ | $0.0318 \pm 0.0107$ |

2012). The numerical solution is denoted by $q_m$. We ignore the discretization error in $q_m$ (which is on the order of $10^{-5}$) and treat it as the "exact" solution. Then the "exact" solution for the committor function in 10D is given by $q(x) = q_m(x_1, x_2)$, $x \in \Omega = \{x : (x_1, x_2) \in \tilde{\Omega}, x \in \mathbb{R}^{10}\}$. Figure 2 (left panel) shows the contour plots of the $2D$ rugged Mueller potential and the committor function $q_m$ at $k_B T = 10$ (the energy barrier from $A$ to $B$ is about 100).

Next, we compute the committor function in the 10D space using the method proposed in section 3. The Mueller potential (19) has two local minima around $a \approx (-0.558, 1.441)$ and $b \approx (0.623, 0.028)$, respectively. We take the two metastable sets $A$ and $B$ as the cylinders centered at $(x_1, x_2) = a$ and $(x_1, x_2) = b$ respectively, each with radius $r = 0.1$. The function $\chi_A(x)$ is constructed as

$$\chi_A(x) = \frac{1}{2} - \frac{1}{2} \tanh \left[ 1000 \left( |(x_1, x_2) - a|^2 - (r + \delta)^2 \right) \right], \quad x \in \mathbb{R}^{10} \tag{20}$$

with $\delta = 0.02$ and similarly for $\chi_B(x)$. These two functions satisfy (11) approximately.

We generate the data at the artificial temperature $k_B T' = 20$ by solving the Langevin equation (2) using the Euler-Maruyama scheme with time step $\Delta t = 10^{-5}$. We take one sample for every 100 time steps, and only keep those data points with coordinates $x \in \Omega \setminus (A \cup B)$. Of these data, 70% and 30% serve as the training and validation dataset respectively. The neural network used in this example is fully connected, and the hyperbolic tangent (tanh) function is used as the activation function in the hidden layers. We take the batch size as $10^5$ and use the package *Tensorflow* (Abadi et al., 2016) with *Adam optimizer* (Kingma & Ba, 2014) to train the network at the physical temperature $k_B T = 10$.

To quantitatively evaluate our results, we compare the committor function $q_\theta$ obtained using the neural network with the FEM result $q$. For rare events considered here, the committor function near the transition state region (i.e. the region near the minima of $V(x)$ on the iso-surface where $q \approx 1/2$) is of our interest, since this region represents the bottlenecks to the transition. The committor function in this region is also the most difficult to compute, as data here is typically scarce. Therefore, we shall focus our evaluation of the numerical results in this region. To this end, we carry out the constrained sampling in the transition state region by simulating the dynamics

$$\dot{x} = -\nabla \left( V(x) + V_q(x) \right) + \sqrt{2\beta^{-1}}\eta, \tag{21}$$

where the additional potential $V_q(x) = \frac{1}{2}\kappa(q_\theta(x) - \frac{1}{2})^2$ with $\kappa = 3 \times 10^4$ is to constrain the system on the $1/2$-isocommittor surface $\Gamma_{1/2} = \{x \in \Omega : q_\theta(x) = 1/2\}$. After equilibration, we sampled 100 points. These points, projected on the $(x_1, x_2)$ plane, are shown in Figure 2 (middle and right panels). The region where these points are clustered is the transition state region. Also shown in Figure 2 (middle and right panels) is a comparison of the $1/2$-isosurface of $q_\theta$ with that obtained from the FEM calculation. Evidently these two agree well in the transition state region. Certain discrepancy occurs away from the transition state region, due to the lack of training data there; nevertheless, those regions are irrelevant to the transition events.

We also computed the root-mean-square error (RMSE) and mean-absolute error (MAE) between $q_\theta$ and the FEM solution at the sampled 100 transition states. The results are reported in Table 1. The errors are based on 10 independent computations. We observe that the numerical results are insensitive to the number of hidden layers, but the accuracy improves for larger set of training data.

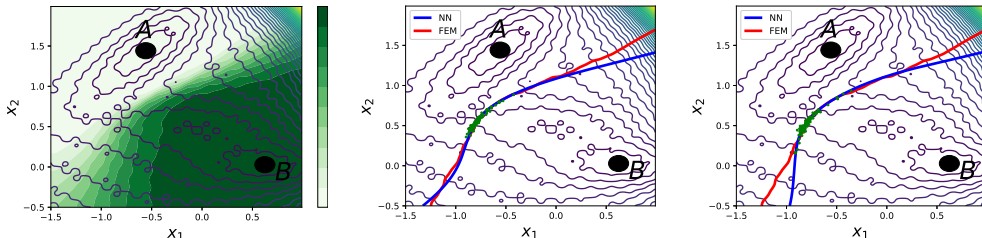

Figure 2: Left: The green filled contours indicate the committor function $q_m$ computed using FEM at $k_B T = 10$. Middle and right: The 1/2-isosurfaces (projected onto the $(x_1, x_2)$ plane) of the committor function $q_\theta$ obtained using the neural network and that obtained from the FEM calculation at $k_B T = 10$. The two panels depict the numerical results using the 10-20-1 network (i.e. 1 hidden layer with 20 nodes; middle panel) and the 10-20-20-1 network (i.e. 2 hidden layers with 20 nodes on each layer; right panel), respectively. The discrete points show the sampled 100 transition states.

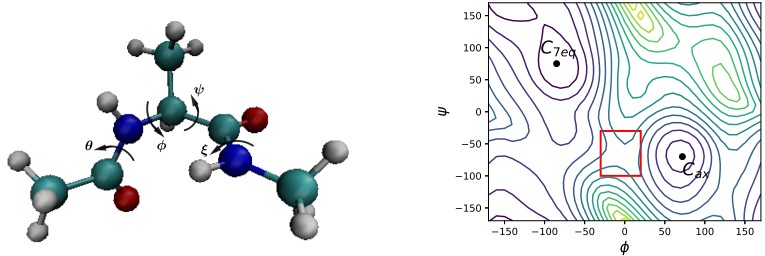

Figure 3: Left: Schematic representation of the alanine dipeptide ($CH_3$-CONH-CHCH$_3$ -CONH-CH$_3$). Right: The adiabatic energy landscape of alanine dipeptide, which is obtained by minimizing the potential energy of the molecule with $(\phi, \psi)$ fixed. The red rectangle indicates the transition state region from earlier studies.

## 4.2 ALANINE DIPEPTIDE

In this example, we study the isomerization process of the alanine dipeptide in vacuum at $T = 300K$. The isomerization of alanine dipeptide has been the subject of several theoretical and computational studies (Apostolakis et al., 1999; Bolhuis et al., 2000; Ma & Dinner, 2005; Ren et al., 2005), therefore it serves as a good benchmark problem for the proposed method.

The molecule has a simple chemical structure, yet it exhibits some of the important features common to biomolecules. Figure 3 shows the stick and ball representation of the molecule (left panel) and its adiabatic energy landscape on the plane of the two backbone torsion angles $\phi$ and $\psi$. There are two metastable conformers $C_{7eq}$ and $C_{ax}$ located at $(-85°, 75°)$ and $(72°, -75°)$, respectively. Accordingly, the metastable sets $A$ and $B$ are chosen as

$$A = \{x : |(\phi(x), \psi(x)) - C_{7eq}| < 10°\}, \quad B = \{x : |(\phi(x), \psi(x)) - C_{ax}| < 10°\}. \quad (22)$$

Our goal is to compute the committor function and then sample the transition states at the temperature $T = 300K$. To generate the training data, we raise the temperature to $T' = 800K$, and use the parallel molecular dynamics package *NAMD* (Phillips et al., 2005) to simulate the Langevin dynamics of the molecule. We run $10^8$ steps with the time step $\Delta t = 2$ fs, and sample one data for every 100 steps. Figure 4 shows the distribution of the sampled data points on the $(\phi, \psi)$ plane (right panel). As a comparison, we also run the dynamics at the physical temperature $T = 300K$, and plot the distribution of the sampled data (left panel). It is seen that at the temperature $T = 300K$, no transition from $C_{7eq}$ to $C_{ax}$ is observed within the simulation time; consequently, no training data is collected in the transition state region, the region of our interest. This demonstrate the benefit of the importance sampling technique proposed in this work.

Next, we discuss feature engineering via collective variables. It is known that, very often the conformational changes of bio-molecules can be adequately described by a few collective variables such as torsion angles. Using this information, we design the first layer of our network to extract 9 such

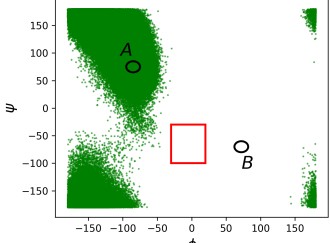 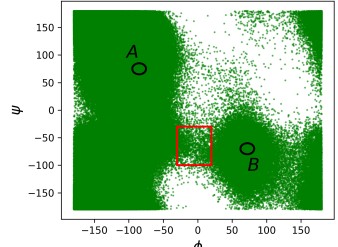

Figure 4: Distribution of the data sampled from the Langevin dynamics of alanine dipeptide at the temperature $T = 300K$ (left) and $T' = 800K$ (right).

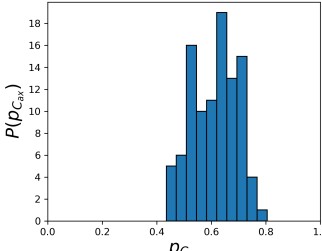 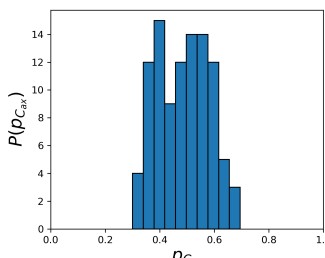

Figure 5: Distribution of committor values for the $100$ states sampled on the $1/2$-isosurface of $q_\theta$. The committor values are computed from direct Monte Carlo simulation. Left: 66-18-30-30-1 network; Right: 66-18-20-20-20-1 network.

torsion angles from the molecule, whose sines and cosines are then fed into the hidden layers as extracted features. Note that these include the four torsion angles identified in Maragliano et al. (2006) as approximately adequate in describing the transition, but we do not a priori assume that we know this precise information, and a redundant description is supplied.

The results shown below were obtained using two million data points sampled outside $A$ and $B$ at $T' = 800K$. These data are equally split into the training and validation sets. We first train the network at $T' = 800K$ by setting the likelihood ratio equal to 1. Then we use the result as the initial network to compute the committor function at $T = 300K$. The training of the network is done by using the *Adam optimizer* with batch size $5 \times 10^5$. The computation is terminated when the validation error no longer decreases.

For this high-dimensional problem, we cannot afford to compute the committor function using the FEM method as we did in the first example. In order to check the accuracy of the numerical results, in particular, whether the $1/2$-isocommittor surface really locates the transition states, we carry out the constrained Langevin dynamics simulation on the $1/2$-isocommittor surface at $T = 300K$. Following the dynamics, we collect 100 states. The committor values of these states are computed directly using the Monte Carlo method. Specifically, we generate 200 trajectories initiating from each of states with random initial velocities and estimate the probabilities (i.e. committor values) of the system reaching $B$ before $A$ (c.f. (4)). If adequate accuracy is achieved, we would expect the computed probabilities to cluster around $1/2$. Figure 5 shows the resulting distribution of the committor values, which confirm the efficiency and accuracy of our method.

## 5 CONCLUSION

In this paper, we introduced a method of computing the committor function at low temperatures, which characterizes rare transition events between metastable states. The main idea is to combine deep learning with importance sampling and feature engineering in order to overcome the curse of dimensionality associated with realistic systems and the scarcity of transition events at low temperatures. Our method is demonstrated to be effective on a relatively simple example involving

the rugged Mueller potential, as well as a more complex benchmark example of the alanine dipeptide molecule. This provides an alternative approach to study complex systems with rough energy landscapes.

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
