# OpenReview forum: "Computing committor functions for the study of rare events using deep learning with importance sampling"
_ICLR.cc/2019/Conference_

### Official Review · AnonReviewer2 · 2018-10-31

**Rating:** 7
**Confidence:** 4

**Review:**

This paper presents a method to train NNs as black box estimators of the commitor function for a physical, statistical mechanical, distribution. This training is performed using samples from the distribution. As the committor function is used to understand transitions between modes of the distribution, it is important that the training samples include points between modes, which are often extremely low probability. To address this concern, this paper draws MCMC samples at a high temperature, and then uses importance weights when training the committor function using these samples. Overall -- this seemed like a good application paper. It applies largely off-the-shelf machine learning techniques to a problem in physics. I don't have enough background to judge the quality of the experimental results.

I had one major concern: the approach in this paper is motivated as a solution to estimating commitor functions in high-d. The variance of importance sampling estimates typically increases exponentially in the dimensionality of the problem, so I suspect this technique as presented would fall apart quickly if pushed to higher dimensions. All experiments are on problems with either 9 or 10 (effective) degrees of freedom, which from the ML perspective at least is quite low dimensional, and which is consistent with this exponentially poor scaling. There are likely fixes to this problem -- e.g. the authors might want to look into annealed importance sampling*.

more specific comments:

"and dislocation dynamics" -> "dislocation dynamics"

"One can easily check" -> "One can check" :P

eq 5 -- this is very sudden deep water! Especially for an ML audience. You should either give more context for the Kolmogorov backward equation, or just drop it. (The Kolmogorov formulation of the problem is not used later, and for an ML audience describing the task in terms of it will confuse rather than clarify.)
what is \Delta q? Does that indicate the Laplacian? Not standard ML notation -- define.

similarly, define what is intended by \partial A and \partial B (boundary of the respective regions?)

eq. 9 -- nit -- recommend using a symbol other than rho for regularization coefficient. visually resembles p, and is rarely used this way. lambda is very common.

eqs 10/11 -- include some text motivation for why the definition of chi explicitly excludes the regions inside A and B.

eq 14: cleverly formulated!

eq 14 / eq 20:
factor of 1000 is very fast! corresponds to an epsilon of O(1e-3). You need to make sure that training samples are generated in the epsilon width border around A and B, otherwise the effect of chi will be invisible when training q_theta. So it seems like epsilon should be chosen significantly larger than this. Might want to include some discussion of how to choose epsilon.

* Totally incidental to the current context, but fascinatingly, annealed importance sampling turns out to be equivalent to the Jarzynski equality in nonequilibrium physics.

---

> ### Author Response · Authors · 2018-11-09
> **Reply to Reviewer 2**
>
> We thank the reviewer for comments and constructive suggestions. Below we answer some of the reviewers questions.
>
> Major concern: As we understand, the annealed importance sampling deals with the issue of singularity of measures, i.e. when the importance sampling distribution and the target distribution may not be absolutely continuous, i.e. their support do not overlap well. The reviewer rightly points out that this can occur, but we note that this occurs in our case not when the dimension is high, but when the inverse temperature difference $\beta-\beta’$ is large. For the temperatures chosen in the currently application, this does not occur, but if we were to consider calculating the committor function at a much lower temperature, still using the same high temperature samples, then we would run into this problem. The annealed sampling technique (also parallel tempering) is definitely useful in this case to make use of a sequence of interpolating importance sampling distributions.
>
>
> Eq. 5: We actually needed this for the finite-element calculation in the first example to quantitatively measure the neural net approximation errors. Thus, we leave this equation here, but we add a reference and more notation clarifications in the revision.
>
> Eq 10/11: the problem is actually defined only in the region excluding A and B, because the definition of q(x) involves hitting times of the boundaries of A and B (see Eq. 4). We made this more clear in a revision.
>
> Eq 14 / Eq 20: Note that the width of the “transition layer” of chi (with factor 1000) is about 0.04 (in x scale, not x^2 scale). This is acceptable numerically. We have tried larger epsilon (lower factors in Tanh). The accuracy does decay a little and we should pick a epsilon that is not too large. Picking epsilon too small causes computational issues. So far we picked it empirically.
>
> Eq. 20: we fixed a typo missing a delta term.
>
> Other typos are fixed and suggested notation changes are implemented.

---

> > ### Comment · AnonReviewer2 · 2018-11-09
> > **exponential increase in variance of importance weights with dimension**
> >
> > EDIT: Fixed a sign error in code below.
> >
> > For a given fixed temperature difference, the variance of the importance weights typically increases exponentially in the effective dimensionality of the problem. This is probably faster+easier to show with a toy experiment than a long-form text explanation.
> >
> > Below is a code snippet computing the variance of importance weights where the target is a quadratic well at temperature 1 and the samples come from a quadratic well at temperature 2 (ie, Gaussian distributions with sigma^2 = 1/2 and sigma^2 = 1). The generated plot can be found here:
> > https://drive.google.com/open?id=157NCM7WYsior2kRmWASZogbj4Hp8wirS
> >
> > --
> >
> > import numpy as np
> > import matplotlib.pylab as plt
> > plt.rcParams['axes.facecolor'] = 'white'
> > plt.rcParams['grid.color'] = 'gray'
> > plt.rcParams['grid.linestyle'] = ':'
> > import tensorflow as tf
> > tf.enable_eager_execution()
> >
> > maxd = 64
> > repeats = 100
> > m = 0
> > n = 0
> > for ii in range(repeats):
> >   print ii,
> >   X = tf.random.normal((1000000,maxd)) # samples at T=2
> >   logp = lambda X, sigma2: -X**2 / (2 * sigma2) - 0.5*tf.log(sigma2*2.*np.pi)
> >   logw_per_dimension = logp(X, 0.5) - logp(X, 1)
> >   logw_cum = tf.cumsum(logw_per_dimension, axis=1)
> >   w_cum = tf.exp(logw_cum)
> >   m += tf.reduce_mean(w_cum, axis=0)/repeats
> >   n += tf.reduce_mean(w_cum**2, axis=0)/repeats
> > v = n - m**2
> > plt.semilogy(np.arange(maxd)+1, v)
> > plt.xlabel('dimensionality')
> > plt.ylabel('variance')
> > plt.title('Variance in importance weights vs. problem dimensionality,\nfor quadratic wells at T=2 and T=1')

---

> > > ### Author Response · Authors · 2018-11-22
> > > **Reply to Reviewer 2**
> > >
> > > We thank the reviewer for the interesting discussion.
> > >
> > > We agree with the reviewer that the variance of the normalized importance ratio will grow exponentially with dimensionality for the case of independent product measures.
> > > Indeed, this variance has a significant effect on the calculation of expectation using importance sampling.
> > >
> > > However, for our variational problem we are not directly computing an expectation, but we are computing the minimizer q. Moreover, the importance ratio does not explicitly depend on q. Hence, it remains unclear how much the variance of the ratio affects the efficiency of the minimization algorithm to find the minimizer.
> > >
> > > In any case, we agree that the importance sampling algorithm can be further improved using annealed importance sampling and related methods.

---

### Official Review · AnonReviewer3 · 2018-11-08
**summary**

**Rating:** 5
**Confidence:** 4

**Review:**

The paper proposes to apply neural network to compute the committor function arose from physics, which looks an interesting application problem by employing machine learning algorithms. Typically, I know very well that the BG distribution usually has multi-modes which makes the sampling difficult extremely. The authors then employ the importance sampling for possibly explore the whole variable space. It seems to me the only possible contribution is parameterize the committor function by using a neural network.
The committor function is parametrized by using a neural network. My first concern is the training data. How would you collect the training data? It is well-known that a neural network works best when there are plenty of training data. Presumably when you are collecting the data, you are basically calculate the committor function and so that you may be able to directly solve the variational problem.
Importance sampling: I would not consider the importance sampling as a big deal concerning the contribution of the paper. You could employ a series of importance distributions which could result in many more samples. Have you also looked at the Uniform distribution?
The paper targets high-dimensional problem. However, in the experiments, the problems do not look really like high-dimensional problems.
Some notations need to be clarified, for example, the Nabla, the Delta, and as well as the dot operator.

---

> ### Author Response · Authors · 2018-11-09
> **Some clarifications**
>
> We thank the reviewer for the comments, and we would like to clarify a few important misconceptions that the reviewer has regarding our work.
>
> “It seems to me the only possible contribution is parameterize the committor function by using a neural network.”
>
> This is not the only contribution. Our contribution is the parameterization of the committor function (with the form (14) to enforce boundary conditions), the importance sampling method and the neural network architecture itself, which is not a vanilla feed-forward network and has a special input layer extracting angles.
>
> “My first concern is the training data. How would you collect the training data? It is well-known that a neural network works best when there are plenty of training data. Presumably when you are collecting the data, you are basically calculate the committor function and so that you may be able to directly solve the variational problem.”
>
> The training data is collected by integrating the Langevin dynamics (2), but at a higher temperature 1/\beta’ than the actual temperature 1/\beta of the committor function we are trying to compute.
>
> First, it is *not* true that integrating the Langevin dynamics is equivalent to solving for the committor function. We use a long trajectory of Langevin dynamics to train our network. To solve for the committor function itself using Monte Carlo samples from Langevin dynamics, one requires, at *each* point in space, to launch several (100-200) trajectories and compute the hitting time probabilities as defined by equation (4). Moreover, without knowledge of the transition region, one has to compute the committor value *everywhere*. Hence, these are not equivalent and the latter is prohibitively expensive. In our case, by sampling a long trajectory, we can estimate p_\beta and hence solve the variational problem with the neural network acting as an approximator. This is much more efficient.
>
> Second, all our sampling is done at a higher temperature \beta’, and the committor function is computed at a lower temperature \beta. Hence, not only is it inefficient to compute the committor function directly from Langevin dynamics, in our case it is not possible, because the data are from a higher temperature. This is a crucial difference and one of the main innovation of this work. As we show in figure 4, if we sample from the low temperature dynamics itself, there would be no points in the transition region, which is the region of interest for the committor function.
>
> “Importance sampling: I would not consider the importance sampling as a big deal concerning the contribution of the paper. You could employ a series of importance distributions which could result in many more samples. Have you also looked at the Uniform distribution?”
>
> First, as discussed before, sampling at a higher temperature is very important because since at the original temperature, there would not be *any* transition data available to train the neural network (Fig 4). Our sampling procedure allows us to use data from a higher temperature to solve for the committor function at a lower temperature.
>
> The choice of our importance sampling distribution (same form as original measure, but at a higher temperature) is also important and not arbitrary. Uniform distribution (or some arbitrary hand-crafted importance distribution) is not appropriate here for the following reason: our goal is to place as much data as possible in the transition region (which is a priori unknown!), and the Langevin dynamics at this higher temperature does this for us. It is extremely inefficient to cover the whole space using uniform distribution for high dimensional spaces (~10), in the hopes of landing some data in the unknown transition region. Consequently, most of the sampled points will be wasted since we are really interested in the transition region only. This gets worse as dimensions increase.
>
> “The paper targets high-dimensional problem. However, in the experiments, the problems do not look really like high-dimensional problems.”
>
> While we agree that 10-66 dimensional problems are not “high” by machine learning standards, in numerical analysis this is quite high. For example, although we could directly solve for the committor function by solving the Backward Kolmogorov Equation for the Mueller potential (can be reduced to a 2D problem) using the finite element method, this becomes intractable for the Alanine dipeptide problem. Hence, we are addressing a high dimensional problem in this application domain. Nevertheless, we believe that these proposed methods do generalize to larger dimensions.
>
> We will incorporate some of these clarifications into the paper in a revision very soon so these confusions are less likely to arise.

---

### Official Review · AnonReviewer4 · 2018-11-10
**Experiments need to be improved**

**Rating:** 6
**Confidence:** 4

**Review:**

In response to the authors' rebuttal, I have increased my ratings accordingly. I strongly encourage the authors to include those ablative study results in the work. I also strongly recommend an ablative study on importance sampling so as to provide more quantitative results, in addition to Fig. 4. Finally, I hope the authors can consider more advanced importance sampling techniques and explore whether it helps you get better results in even higher dimensions.

=================================
This paper proposes several enhancements to a neural network method for computing committor functions so that it can perform better on rare events in high-dimensional space. The basic idea is using a variational formulation with Dirichlet-like boundary conditions to learn a neural committor function. The authors claim to improve a previous neural network based method by i) using a clever parameterization of the neural committor function so that it approximately satisfy the boundary condition; ii) bypassing the difficulty of rare events using importance sampling; and iii) using collective variables as feature engineering.

Generally I feel this paper is well written and easy to understand, without requiring too much background in physics and chemistry. The application is new to most people in the machine learning community. However,
the main contributions of this paper are empirical, and I found the experiments not very convincing. Here are my main concerns:

1. There is almost no ablation study. The parameterization of committor function satisfies the Dirichlet boundary condition, which is aesthetically pleasing. However, it's unclear how much this improves the regularization used in the previous method. Similarly, without importance sampling, will the results actually become worse? What changes if the collective variables are removed? There is even no comparison with the previous neural network based method on computing committor functions, though the authors cited it.

2. In the experiment on extended Mueller potentials, authors use the FEM results as the ground truth. However, it is not clear how accurate those FEM solutions are. Without this being clarified, it is unclear to me that the RMSE and MAE results in Table 1 are meaningful. Maybe try some simpler problem where the committor functions can be computed exactly?

3. In experiments the authors often argue that results will improve when networks become deeper. However, all network architectures used in the paper are narrow and shallow when viewed from the perspective of modern deep learning. If the authors want to stress this point, I would expect to see more experimental results on neural network architectures, where you vary the depth of the network and report the change of results.

4. "Then we use the result as the initial network to compute the committor function at T = 300K" => Did you first train a neural committor on samples of T = 800K and use its weights as initialization to the neural committor for T = 300K? Please clarify this more.

5. Finally, I think the importance sampling technique proposed in this paper can be improved by other methods, such as annealed importance sampling. The largest dimension tested in this paper is only 66, which is still fairly small in machine learning, and I don't expect the vanilla importance sampling can work in higher dimensions.

---

> ### Author Response · Authors · 2018-11-22
> **Reply to Reviewer 4**
>
> We thank the reviewer for the comments. Below we answer the reviewer questions.
>
> 1. The current method to deal with the Dirichlet boundary condition is simple and easy to implement. In the previous penalty method, training data is needed on the boundaries of the sets A and B. Sampling of these data requires additional work. This problem does not exist in the current method. Moreover, the current method does not require the prescription of the penalty parameter.
>
> We compared the two methods for the 10D Mueller potential at the temperature k_B T=20. The same data sampled from Langevin dynamics are used for both methods; additional data at the boundaries of A and B are also sampled for the earlier penalty method. The results can be found at
>              https://drive.google.com/drive/folders/1UV7H3AQj-hdf-oT96RG6ifD3iLsBhPBw?usp=sharing
>
> The current work is concerned with rare events. A Direct simulation of the dynamics at the physical temperature is inefficient for sampling transition events. We believe this has been made clear and adequate evidence (in particular, Fig 4) has been provided in the paper.
>
> The computation becomes more expensive if the layer of collective variables is removed. The purpose of the feature layer is to save computational cost by incorporating certain a priori knowledge in the neural network structure.
>
> 2. In scientific computing, when the exact solution is not available, it is a common practice to assess the accuracy of a numerical solution using a more accurate solution obtained with finer mesh. We followed a similar approach here. We have checked that the error of the solution obtained using the finite element method is about 10^(-5), which is way below that can be achieved by the deep learning approach. This justifies it being used as the "exact" solution in the assessment of the neural network solution.
>
> 3. For the 10D Mueller potential, we carried out 3 calculations with 1, 2 and 3 hidden layers, respectively. We found the change in accuracy is within statistical errors, and hence not significant. The addition of more data points however, significantly affects accuracy. We have revised the paper accordingly with more thorough numerical evaluations.
>
> 4. Yes. As mentioned in the paper, we first train the network at T’=800K then use the result as initial value to train the network at T=300K.
>
> 5.We agree the importance sampling method in the current work can be improved - no method is perfect. The main difficulty in the study of rare events arises from the disparity of the energy barrier and the effective thermal energy. The purpose of our importance sampling method is to overcome this difficulty.

---

### Official Review · AnonReviewer5 · 2018-11-12
**Interesting application paper, somewhat incremental improvement over prior work**

**Rating:** 6
**Confidence:** 4

**Review:**

This paper looks at the problem of computing committor functions, which is defined as the probability a state first visits a local minimum of the energy landscape in Langevin dynamics. The authors motivates the problem well by explaining why this is difficult to compute. Khoo et al, 2018 uses a deep network to variationally approximate this function. The major contribution of this paper is several improvements to the techniques of Khoo et al. I will comment on each of the improvements in turn.

In section 3.1 the authors proposes that instead of using optimization to satisfy boundary condition, it might be better to parameterize the function to satisfy the boundary condition. This contribution seems incremental. Eq.(9) guarantees satisfaction of the boundary conditions when lambda is large enough, so I imagine that lambda is not very difficult to pick. Therefore, the practical reason to use the more sophisticated parameterization is unclear. The new proposal removes a hyper-parameter lambda, but introduces new hyper-parameters epsilon and the exact smoothing for the two functions X_A, X_B.

The contribution in 3.2 seems interesting. The authors replace the original sampling function, which has high variance, with importance sampling. It seems that importance sampling is very well suited for this problem, and the authors found a very natural and reasonable proposal distribution. This method is generally interesting for estimating the expectation of any random variable with respect to a Boltzmann distribution.

In section 3.3 the authors proposes to work on a feature space. This is interesting to audience interested in the specific applications. But for machine learning this is a standard procedure, so has limited methodology novelty.

One issue of this paper is limited audience in ICLR. It seems much more appropriate to submit to statistical physics, material science or other relevant communities. I am not capable to judge the significance of this paper to those communities. As an application paper, the proposed methods are somewhat incremental; I only find section 3.2 interesting to a broader audience.

Writing:
I like the writing. Everything symbol is defined before use, and the notation is clean and unambiguous. I can easily follow the author’s arguments down to the minor details.

A minor improvement to section 2 is to first explain the shortcoming of Eq.(5), then introduce Eq.(6)(7).

Eq.(11) can be better explained. The definition doesn’t look like a smooth function, and it takes some time to figure out what the authors mean here.

Minor comments:
A, B \in \Omega should be A, B \subset \Omega in Section 2

---

> ### Author Response · Authors · 2018-11-22
> **Reply to Reviewer 5**
>
> We thank the reviewer for the comments.
>
> To handle the boundary condition, the previous regularization method needs to sample additional data on the boundary of A and B. Sampling of these data requires additional work. This problem does not exist in the current method. Moreover, we compared the two methods for the 10D Mueller potential at the temperature k_B T=20. The same data sampled from Langevin dynamics are used for both methods; additional data at the boundaries of A and B are also sampled for the earlier penalty method. The results can be found at
>              https://drive.google.com/drive/folders/1UV7H3AQj-hdf-oT96RG6ifD3iLsBhPBw?usp=sharing
>
> Eq 5 is equivalent to the variational formulation in (6) and (7). However, solving the Kolmogorov equation is not tractable in high dimensions. This is explained in the paragraph after eq (7).
>
> Eq 11 admits functions which vary smoothly from 1 on the boundary of A or B to 0 in a region of width \epsilon. This is explained in the paragraph after eq (12).
>
> We have fixed other typos pointed out by the reviewer.

---

### Meta-Review · Area_Chair1 · 2018-12-13
**Borderline paper**

**Confidence:** 3
**Recommendation:** Reject

**Metareview:**

This paper proposes a neural network based method for computing committor functions, which are used to understand transitions between stable states in complex systems.
The authors improve over the techniques of Khoo et al. with a method to approximately satisfy boundary conditions and an importance sampling method to deal with rare events.
This is a good application paper, introducing a new application to the ML audience, but the technical novelty is a bit limited. The reviewers see value in the paper, however scaling w.r.t. dimensionality appears to be an issue with this approach.